# Digitalization in Just-In-Time Approach as a Sustainable Solution for Maritime Logistics in the Baltic Sea Region

**Olena de Andres Gonzalez** [1,*]**, Heikki Koivisto** [2]**, Jari M. Mustonen** [1] **and Minna M. Keinänen-Toivola** [3]

1. Faculty of Technology, Satakunta University of Applied Sciences, FI-28130 Pori, Finland; jari.m.mustonen@samk.fi
2. Faculty of Logistics and Maritime Technology, Satakunta University of Applied Sciences, FI-26100 Rauma, Finland; heikki.koivisto@samk.fi
3. Faculty of Technology, Satakunta University of Applied Sciences, FI-26100 Rauma, Finland; minna.keinanen-toivola@samk.fi
* Correspondence: olena.de.andres.gonzalez@samk.fi; Tel.: +358-44-710-3239

**Abstract:** This research provides an overview of the process and results of the development and implementation of the Port Activity Application. The aim of the application is to improve the coordination and information exchange mechanisms between the existing systems of ports and ships during piloting ordering process to ensure their effective interoperability, giving a better understanding of the impact of digitalization on the sustainability of seaports and maritime transport. To implement this concept, a system of step-by-step actions was developed, including determining the current situation, developing a business model and business logic for implementing an appropriate information and communication technology (ICT) solution, analysing the local government structure, analysing intermodal information exchange between stakeholders, developing and testing a new ICT tool. The ports of Rauma, Finland, and Gävle, Sweden, were used as pilot ports. As a result of the study, the main bottlenecks in the process of information exchange in ordering pilotage were identified. An improved business model and business logic, that allows the rational use of resources and reduces $CO_2$ emission and the pressure on the environment, was developed. The testbed was conducted in an environment of real port operations. Currently, the open access source code is available for use for maritime cluster actors.

**Keywords:** maritime logistics; sea traffic management; shipping; sustainability; sustainable mobility; port digitalization; just-in-time approach; business model; environment; $CO_2$ emission

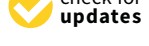



## 1. Introduction

According to the United Nations established sustainable development goals aim at reducing the negative effects of climate change. Attention is especially drawn to the need of reducing emissions of various pollutants in oceans and coastal waters, which includes $CO_2$ and $SO_x$ emissions [1]. In the Baltic Sea region the regulations under International Maritime Organization's (IMO) International Convention for the Prevention of Pollution from Ships (MARPOL) has the most demanding requirements of various emissions, especially for SOx since 2005 [2].

Sea transport plays an important role in world trade. According to the IMO, sea transport covers almost 90% of world trade, and short distance transport accounts for 70% of all trade transport in the world. At the same time, the Baltic Sea region is one of the busiest in the world and Finland, which is geographically almost an island and carries out 90% of export and 80% of import operations through maritime logistics, is a maritime country dependent on maritime transport [3]. According to the study, "Maritime Transport in the Gulf of Bothnia 2030", the sea trade in this part of the Central Baltic Region is expected to grow from 30% (modest forecast) to 60% (strong growth forecast) by 2030 [4].

The Finnish Maritime Transport Strategy and the Baltic Sea Region Strategy are based on sustainability, water sources protection and maritime safety objectives [5].

To service such a flow of goods, ports are involved in the logistics chain as the main location for loading and unloading the ships, as well as terminal operators, port authorities, local authorities, pilots, tug operators, stevedores, waste disposal companies and many others. The workload of ports and port operators leads to a peculiar behaviour of merchant ships, as they strive to get to the port as quickly as possible. This often leads to inappropriate fuel consumption and an increase in the waiting time until the necessary facilities for loading or unloading the ship becomes free and, accordingly, increases $CO_2$ emission and the load on the environment.

E-navigation and just-in-time operations, resulting in emission control and promotion of sustainable development, are considered by IMO to be a major challenge for modern shipping [2]. Electronic navigation is designed to provide digital information and infrastructure for maritime safety, protection of the marine environment, reducing administrative burdens and increasing the efficiency of maritime trade and transport [6]. Taking their lead from the vision of intergovernmental organizations, such as the IMO and the EU, other intergovernmental and international organizations are also proceeding with detailed implementations to improve data sharing and effectiveness in the maritime transportation chain.

This paper explores the first implementing conception of STM (Sea Traffic Management) in the Baltic Sea region and just-in-time operations during piloting ordering process with particular emphasis on its impact on sustainability in the Baltic Sea region. Sea Traffic Management (STM) focuses on efficient, secure and environmentally sustainable sea transports from berth-to-berth [7]. STM includes Port Call Synchronization—the continuous process of coordinating a ship's approach, its previous and next ports' operations, hinterland transport operators' plans, and the progress of a port call; and Port Call Optimization—the sharing by port call actors their estimated and actual times regarding certain states in the port call process. The research problem stems from an increase in the load on the environment, and an increase in greenhouse gas (GHG) and $CO_2$ emissions as a result of irrational planning of the arrival time of the vessel at the port.

The aim of the article is to describe the use of a just-in-time approach and the implementation of the STM concept by testing a developed ICT tool, in terms of economic and environmental aspects of sustainability for the port ecosystem in the Baltic Sea region. The Electronic Data Interchange (EDI) systems currently in use need to be optimized as nowadays the flow of goods and information is very intensive [8]. In order to increase the efficiency of the daily operations, all participants regardless of their place in the logistics chain should have access to reliable and up-to-date information.

## 2. Materials and Methods

The possibilities for improving the maritime transport quality and options for just-in-time arrivals and departures between Finland and Sweden were investigated, in order to optimize the flow of goods by sea, and to better integrate sea transport with road and rail transport. As a case study, the results of Interreg Central Baltic EfficientFlow project are presented. EfficientFlow project was a joint Swedish-Finnish initiative that contributed to the development of two transport corridors in the Central Baltic area, i.e., the corridor to and between the ports of Gävle (Sweden) and Rauma (Finland). The main aim of the three-year EfficientFlow research was to make the transport flow in the corridors more efficient by improved processes, in order to produce new digital solutions and to demonstrate a successful case of the first implementation of the STM concept in the Baltic Sea Region.

### 2.1. Pilot Ports: Port of Gävle and Port of Rauma

Port of Gävle (Gävle, Sweden) and Port of Rauma (Rauma, Finland) were chosen as case ports for EfficientFlow research because of their similarity in size, goods categories, traffic flows and hinterlands. In addition, the ports are located almost on the same

latitude with similar weather conditions. However, despite their similarities, they are not competitors.

The Port of Gävle is located on an inlet of the Gulf of Bothnia in East Sweden, 174 kilometres northwest of Stockholm. Port of Gävle is one of top 10 Swedish ports by volume of goods [9] which handles over 200,000 twenty-foot equivalent units (TEU) and six million tonnes of cargo yearly. Gävle's eight terminals provide services to more than 1000 ship calls per year. Port serves mainly the wood and steel industries, but also liquids for e.g., Stockholm Arlanda Airport. Daily around 20 trains leave the port, carrying goods all around the country. Since 2014 the port has a new and deeper fairway, allowing to host ships with a beam of 42 m and a draught of 12.2 m. Tripling the channel's width made possible to receive ships up to 100,000 GT and eliminated difficulties caused by weather conditions and allowed night calls [10].

Located on the southwest coast of Finland, Port of Rauma is the fourth biggest port in the country, handling almost 300,000 TEU and around six million tons of cargo every year. With its 20 berths the Port of Rauma provides multipurpose facilities for import, export and transit traffic. The main export cargo types, include caring paper, cardboard, pulp, products of the chemical and metal industries; import freight types are raw materials and chemicals for the wood industry. The transport modes are trucks and trains. The Port of Rauma is known for its expertise in handling high and heavy items, project shipments, and for its multipurpose services and safe and flexible operations [11].

The two ports were found to have a lot of congruencies regarding their development strategies, specialization and geographical location. Both ports are geographically located close to the Scandinavian—Mediterranean transport corridor. These transport corridors are an important part of the Trans-European Transport Network (TEN-T) and on their territories lie the majority of the existing Finnish and Swedish mining, metal, forest and other manufacturing industries. One of the most important aspects of being a sustainable port is the port's environmental impact. Therefore, the measurement and the control of the harmful gases should be among the port's top priorities. Port of Rauma has been tracking GHG emissions data since 2005. At the port of Rauma, the harmful emissions decreased from almost 16,000 tonnes per year in 2014, to less than 10,000 tonnes per year in 2018. Port of Gävle has also started collecting GHG emissions data from 2017 onwards.

*2.2. Definition of Current Situation of Port Flow Optimization and Pilot Ordering*

The project research was started by defining the current situation of port flow optimization and pilot ordering and the processes needed for organizing workshops and other activities between the project partners and stakeholders, including: Interview of Port of Rauma and Port of Gävle authorities, interview of port of Rauma's terminal operator, distribution of a questionnaire among stakeholders, comparison of some of the ports located in the Central Baltic Region, and overview of research-related literature [12–24].

Secondly, to define the current situation, a questionnaire was composed and distributed to over 100 companies and organizations cooperating with port of Gävle and port of Rauma. The purpose was to determine if there were: Bottlenecks in the ongoing processes of port flow optimization and pilot ordering; issues in the practices currently in use; needs of a real-time data sharing system with live updates; any recommendations and suggestions. The questionnaire was divided into three parts: Part 1 "Background information" aimed to divide the stakeholders into categories depending on their volume of trade. Each category was evaluated individually and against an average of the others. This determined what the areas of interest of the different participants are and how open they are towards implementation of innovations. Part 2 "Information and Goods Handling" gave the opportunity to compare the current Electronic Data Interchange (EDI) systems with the ICT tool. Part 3 "Service Evaluation" represented a brief customer satisfaction feedback form.

Based on the questionnaire (SMA, Sweden), which is presented in more detail in the third section, "Results", of this paper, the bottlenecks have been found in the piloting order-

ing process. Based on the information received from the workshops and the questionnaire, developing of the Port Activity Application continued with detailed analyses of the steps of the piloting ordering. For identifying the features of piloting ordering, the way of working was investigated in pilot station Bönan (Gävle, Sweden). A systems overview was made with a several IT systems used when ordering and planning pilotage. As the investigation only covered some actors of the field, it was acknowledged that ways of working at pilot stations in Sweden and Finland can vary.

### 2.3. Business Model and Business Logic

The scope of the analysis on the business model and the business logic currently in use in the ports was to introduce an optimized and more efficient business model. It was made by conducting a thorough analysis of the business model and the business logic currently in use in the ports, between the port call actors, including the logistics chain. For this purpose, the ports of Rauma and Gävle were used as a pilot ports.

The current business model and business logic were determined by carrying out the following activities: Partner meetings and multiple workshops between the partners and stakeholders in Finland and Sweden; series of interviews of port call actors: Terminal operators, pilots, tugboat companies, shipping line, customs and hinterland operators. In addition, reviewing of literature was carried out [4,25–37].

The analysis of the outcome of the above-mentioned tasks revealed how the enhanced business model and enhanced business logic provided the desired time and resource savings without compromising on safety and environment. The business canvas tool was used for presenting the enhanced business model and business tools.

### 2.4. Establishment of Local Governance Structure

Establishment of local governance structure aimed at creating a platform for collaboration between port call actors involved in the port call process. Local governance structure described a framework for exchanging information. It played a vital role in supporting the collaboration between the different actors by assigning them suitable roles. By accepting the role, each actor accepts certain duties and responsibilities. It had to be considered that the different partners may have different priorities. Therefore, the local governance's vision, mission and strategic objectives had to be adequately communicated, well understood and accepted by all partners in order to avoid any partnership disturbances and governance issues.

The key partners were identified for the establishing the local governance structure with information on operations, their purpose and values as well as their contribution to the logistic chain for both ports, the Port of Gävle and the Port of Rauma. The local governance structure identification was based on the previous step of analysis of the business model and the business logic and following activities: Joint workshops together with partners and stakeholders; interviews of partners and stakeholders; literature and regulatory reviews [38–46].

### 2.5. Information Flows for Intermodal Sharing of Information

As the Port Activity Application developed by the project is an important achievement in the digitalization of the port operations, the research was looking ahead presenting port digitalization concept and intermodal information sharing between the key stakeholders of port and hinterland logistics operators. The research was started with desktop work and a literature review [10,11,47–51]. Several live and recorded webinar presentations and white papers on related topics were visited to grasp the latest developments in the field.

The main approach focused on the information flows between the port operators and relevant hinterland logistics operators (trucks and trains). To illustrate this, a stakeholder Metromap, including hinterland logistics operators was developed. The Metromap highlights a detailed scheme of the interaction of all stakeholders in the intermodal transportation process. Intermodal transportation is a combination of two or more transportation

modes within the logistic chain from the shipper to the consignee. For example, a combination of ship, train, and truck transportation. Also, the forwarder or transportation agent is different from stage to stage. To allow efficient handover from a transportation mode to another, a standardized package is a necessity. Industry standard is to use cargo containers with standardized physical properties. Intermodal transport can be characterized by optimized frequency of services and by the freedom of choosing the most economically and logistically suitable transportation mode. These define reliable and timely delivery resulting in improvements in productivity, inventory, cycle times and economics [10,11,47–51]. The Metromap proved to be useful when explaining the interconnection between the port and hinterland in the auditions where EfficientFlow Port Activity App presentation was used.

Initially, the timely target was to conduct the key stakeholder qualitative interviews during springtime 2020 as face-to-face meetings and discussions. Due to the COVID-19-epidemic outbreak, this plan was however put on hold and the semi-structured interviews were eventually held online. Finally, it was decided to make the presentations and have discussions with relevant and interested hinterland stakeholders during May and June 2020.

A series of interviews with hinterland logistics operators was conducted to gain understanding on what information is needed and how this information is dealt with. These interviews followed the same basic structure to maintain comparability and simultaneously allowing flexibility in the discussion to let brainstorming-like ideas to emerge.

The basic structure of the presentation and audition was as follows: Presentation of the main project; presentation of the hinterlands work package; the Metromap presentation; Port Activity Application; timestamps. In addition, the audition included discussion which offered the possibility to ask questions such as: Are you interested in the new application? What timestamps do you find useful? Any further timestamp information which could be useful? Any hinterland timestamps available which could be provided to the port app timestamps? What kind of information/location data/timestamps would you believe should be added to aid your day-to-day operations and streamline your logistic operations?

*2.6. Testbed*

The purpose of the application was to share information and timestamps (estimates and actual) during port calls and put the information into a user-friendly user interface. The data were collected from various sources, port actors and systems, such as Automatic Identification System of vessels (e.g., "Live" in Sweden), Terminal Operator System ("Opera" Finland, Rauma), Port Info System, Pilot System and manual input. The developed Port Activity App was the first implementation of the Sea Traffic Management (STM) concept at the Baltic Sea Region, so extensive testing was expected to be needed. The Port of Rauma, the Port of Gävle and Satakunta University of Applied Sciences (SAMK) experts and students were the first testing groups and after corrections other port stakeholders joined the testing. Due to the feedback from stakeholders of the ports, new features were ordered as additional orders. Various tests were initiated for both applications by several actors (students and experts of Satakunta University of Applied Sciences (SAMK), port actors) to do a complete and thorough examination, reflecting real-life situations. Test methods used in testbed of Port Activity Application were: (1) Usability testing for the user interface of the applications; (2) integration testing for validating the data from different sources; (3) system testing to identify server capabilities for multiple users at the same time; (4) security testing to see the application's General Data Protection Regulation (GDPR) compliance; (5) bug-hunting for finding the most critical bugs and malfunctions.

The first user test of the Port Activity App was made with SAMK (Pori, Finland) student groups to discover most critical bugs the application has this far and to investigate user interface (UI) and user experience (UX).

The selected SAMK experts did practical user testing by using the Port Activity apps and verified their proper function. Emphasis was on validating that the data shown by the app was correct and up to date. This was done by comparing the app data with the data of other available on-line sources like Marine Traffic [52], Finnpilot [53] and Fintraffic [51].

First user access rights were given to testing experts so that they were able to fully use the app and all its features. Ships visiting Port of Rauma regularly were divided into fleets, one for each tester. This was done to avoid overlapping in error reporting. The tester followed the movement and especially the port calls of the vessels of his/her fleet. Testing tasks or use cases related to port call included arrivals and departures, berthing and pilotage.

The Port of Rauma personnel test group, which included actual port actors, started using the application in their everyday work after tests with students and experts of SAMK were made. During the testing period, weekly meetings were held, and the personnel involved in testing process was continuously reporting to the testing group. Plans for the Port of Rauma test included port actors and have them use the Port Activity Application in their work, performing port calls, checking timestamps and comparing data streams from different port operation systems. Logistics module test was done with the trucking company. The main thing that the Port Activity Application was supposed to do, was to decrease the amount of phone calls during the port calls. Port actors were asked to check the timestamps frequently and to report them if errors occurred. Errors were collected and forwarded to the software developing team for fixing.

The Port of Gävle testing was done with 20 port actors who were testing the Port Activity Gävle application in real-life situations. They had received a test template, where they were able to add comments and report errors. Port of Gävle founded a Customer Advisory Board group (CAB-group) early in the EfficientFlow project. The intention of the CAB-group was to gather dedicated port actors and encourage them to be engaged in the EfficientFlow project in general and in the Port Activity App in particular. Tasks were planned to reflect the real-life use of the application such as logging into the application frequently, checking the data and comparing it to other port operation systems and checking and manually adding timestamps into the application. The testers were asked to use the application for the port calls that the port actors are involved in, to connect it to real-life situations. The results of the tests were utilized to make both applications perform better and to be more user-friendly, for fixing critical bugs and to validate data from different sources and making sure the data was up to date.

In order to define the current situation more in detail, a questionnaire was carried out in both ports. The first questionnaire was used for port actors in Port of Gävle in 2018. The focus of the questionnaire was to find out how much the port actors used phone, e-mail, etc. as communication tools during a port call. The questionnaire of 2018 was followed by a new questionnaire in 2020 after the Port Activity Application was launched and the port actors were able to receive online information through the application. The questionnaires in 2018 and in 2020 were submitted to about 30 port actors. The response rate was little less than 50% both in 2018 and in 2020.

## 3. Results

### 3.1. Definition of Current Situation from Position of Port Flow Optimization and Pilot Ordering

Defining the current situation with information exchange in real time, during piloting ordering between ships and port actors, discovered main bottlenecks of this process. The results of the questionnaire, which was answered by 12 Swedish and 26 Finnish companies, unambiguously showed that most of the issues and bottlenecks were caused by several factors: A lack of information, lack of clarity, use of unreliable and outdated data, poor communication practices, misunderstandings, duplication of work, non-automated processes of data handling, poor or no data sharing, use of obsolete technologies, human factor, unclear roles and responsibilities.

By reviewing the workshop outcomes to define the current situation of port flow optimization and pilot ordering, it was identified that both ports, Gävle and Rauma, needed improved information sharing and support in the challenge of determining estimated times. With Sea Traffic Management (STM) and port call optimization, which the port call actors used to share timestamps between each other, better predictability and improved situational awareness was the desired result of the new ICT tool. Therefore, an urgent need

to implement a common real-time data sharing software interconnecting all participants in the logistic chain regardless of their role was identified. The current practices had to be optimized following the Behaviour Based Safety (BBS) approach in order to meet the current demand and new standards and to automate the process of information handling. Data accessibility and visibility, as well as information reliability and real-time update possibilities were among the first priorities.

*3.2. Business Model and Business Logic*

The analysis of the partner meetings and multiple workshops between the partners and stakeholders as well as the series of interviews with port actors revealed how the enhanced business model and enhanced business logic would provide the desired time and resource savings without compromising on safety and environment. As mentioned in the previous part of this paper, the pilot station Bönan (Gävle, Sweden) was taken as an example for investigation, even though the way of working in other pilot stations in Sweden or Finland had different pilot ordering systems. The Swedish way of ordering a pilot was recommended to be done via the "Maritime Single Window". In some cases, however, the order was made manually over the phone directly to the pilot planner. When communicating with the pilot planner several options were available: phone; email; Maritime Single Window (MSW); marine Very High Frequency radio (VHF). Surprisingly, although there were IT systems available, a lot of the communication was done over the phone or email directly from agent to pilot planner or from pilot to pilot planner. This fact is, of course, a challenge when trying to automate operations. IT Systems used for ordering a pilot were Maritime Single Window (MSW) and Fenix Pilot Planning, Aecdis.

Although the systems currently in use were quite advanced, there were some missing components and links that created bottlenecks and caused waste of time and resources. The front-end users were allowed to use software as PortNet and PortIT, but these systems were not designed according to the users' needs. The main disadvantage was that the various Electronic Data Interchange (EDI) systems were not interconnected in a shared network. An independent standalone system could be characterized as having a good level of customization, relatively higher security and lower service and maintenance cost. However, not being part of a shared network led to the use of outdated data, poor communication practices, misunderstandings and often work duplication. Another consequence of not using a common real data sharing system was the lack of standardization. When the processes of data handling were not automated, the presence of human factor, unidentified roles and unclear responsibilities caused unexpected delays and unwanted situations. In order to satisfy the needs of the whole logistic chain, the software provided coordination among all port actors. The key factor to successful and effective logistics operations was the reliable information exchange between the ships, ports and hinterland operators. The IT systems what were used when ordering and planning pilotage are connected to other systems, but these are left out in this description.

Overall, systems surrounding the pilotage order / pilotage planning were not very automated (Figure 1). Fenix Pilot planning was used more as an advanced notebook with some support for rules, for example, control of pilot license. In most cases the pilotage was already agreed in manual communication when the pilot planner makes actions in the system. Thus, the way of working regarding pilot planning was mostly the same as it has been the last 15–20 years. In the Single Window portal the information was not that well integrated with the consequence as the agent was forced to phone the pilot planning more often than necessary. The technology stack that Fenix Pilot planning and MSW were built on is quite modern and is a good base for further development.

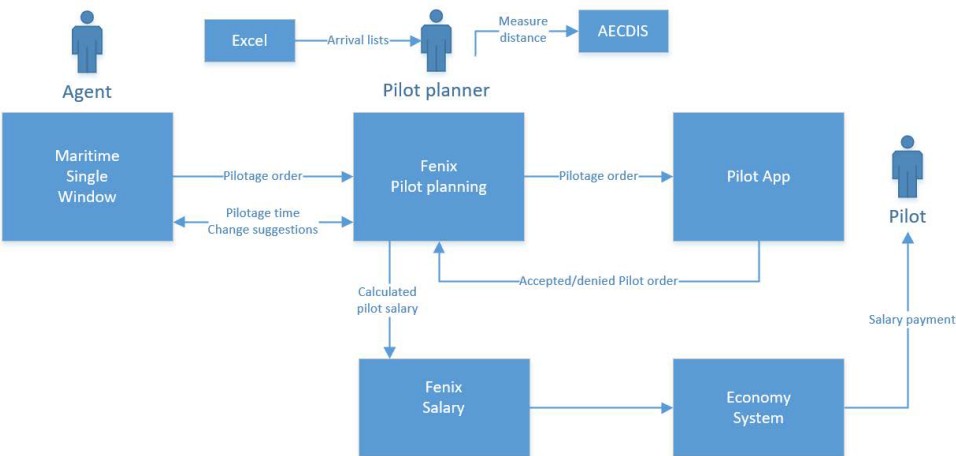

**Figure 1.** Current main used systems, Sweden. Maritime Single Window (MSW) what is based on EU-directive 2010/65/EU [54]. Fenix Pilot Planning is a pilot planner manages ordered pilotage from MSW, enters manual orders, assigns pilotage to pilots, the overall planning. Pilot App is a phone app where the pilot gets the pilot order and can accept/deny order, manage travel bills, see overall pilotages for pilot areas. Fenix Salary is a system used for calculating compensation to pilots based on information from the actual orders. Economy system is a system that keeps track of economical transactions based on information from Fenix Salary. Aecdis is a system used to measure distances.

When developing automatization of the pilot order process, respect was given to all manual actions that were made in the process. For example, when an agent phoned the pilot planner, if the planner was flexible and smooth, a deal was made directly over the phone which meant that the pilotage was moved to when the boatmen already had another connecting voyage planned and the stowage was ready as well. The agent was happy with the answers, no extra fees for anyone, "all are winners". This made automatization even harder, as it often is when moving from a more or less manual process to an automated process.

When the research started, the port call actors were partially connected (Figure 2). However, having internal integration was not enough to meet the higher market demands and the increasing cargo volumes. Even though the participants were linked into a network, the information and knowledge sharing level was very low. The poor data sharing was determined as one of the most important issues in the analysis of the current situation regarding port flow optimization.

The EfficientFlow project focused on the improvement of the "port to port" part of the supply chain. Most of the current IT systems used by different actors were designed independently from each other without following any common standard and requirements. The lack of standardization resulted in issues when system integration and interconnection was needed. This was one of the reasons that limited the implementation of innovative value adding solutions. On the other hand, the traditional communication methods as emails, phone calls and electronic messages created a lot of operational waste. Another issue that must be eliminated or at least minimized is the uneven distribution of the workload. The peak and off-peak periods have to be managed and controlled accordingly as this creates great disturbances in the schedules of shipping lines, hauliers and terminal operators. Most of the time the reason is miscommunicated or delayed information.

The Port Activity Application Metromap, shown in Figure 3, illustrates the complexity, and the need for collaboration between multiple port call actors, in staging a port call. In Figure 3, an example port call was illustrated as a metro map, where each metro line represented an actor and each metro station represents a state, such as the occurrence of events or performance of actions that was important in the port call process. The metaphor illustrated a flow of actions and events that occurred in a port call, from the arrival of a

vessel (left part of Figure 3) to its departure (right part of Figure 3), with terminal operations in between (shown in the dotted ellipsis in the same figure).

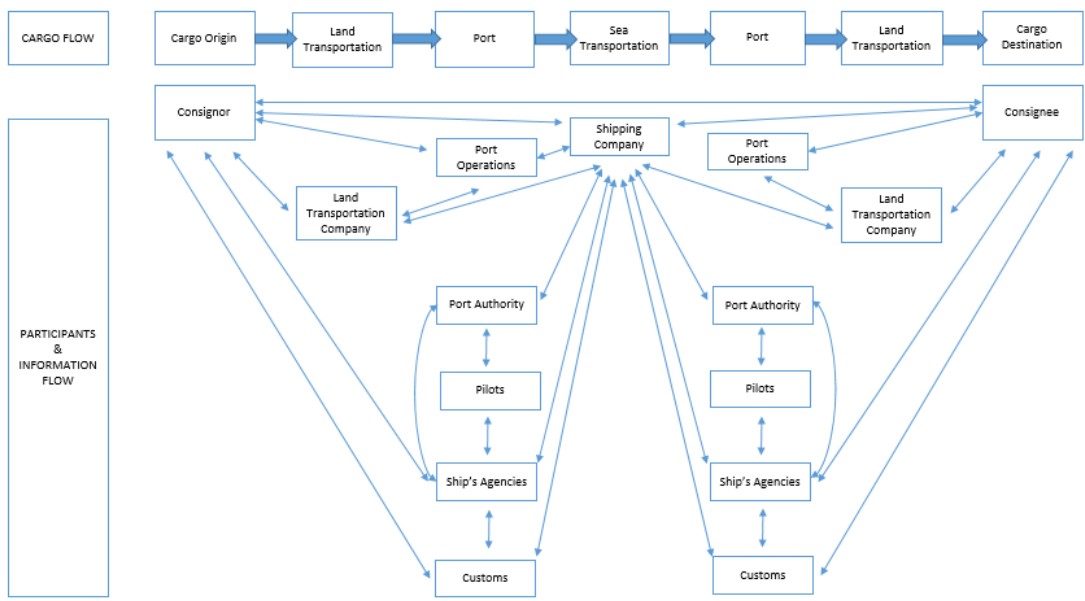

**Figure 2.** Information exchange between stakeholders during the port operations before EfficientFlow research: Data collection for analysing and benchmarking was scattered among the different players.

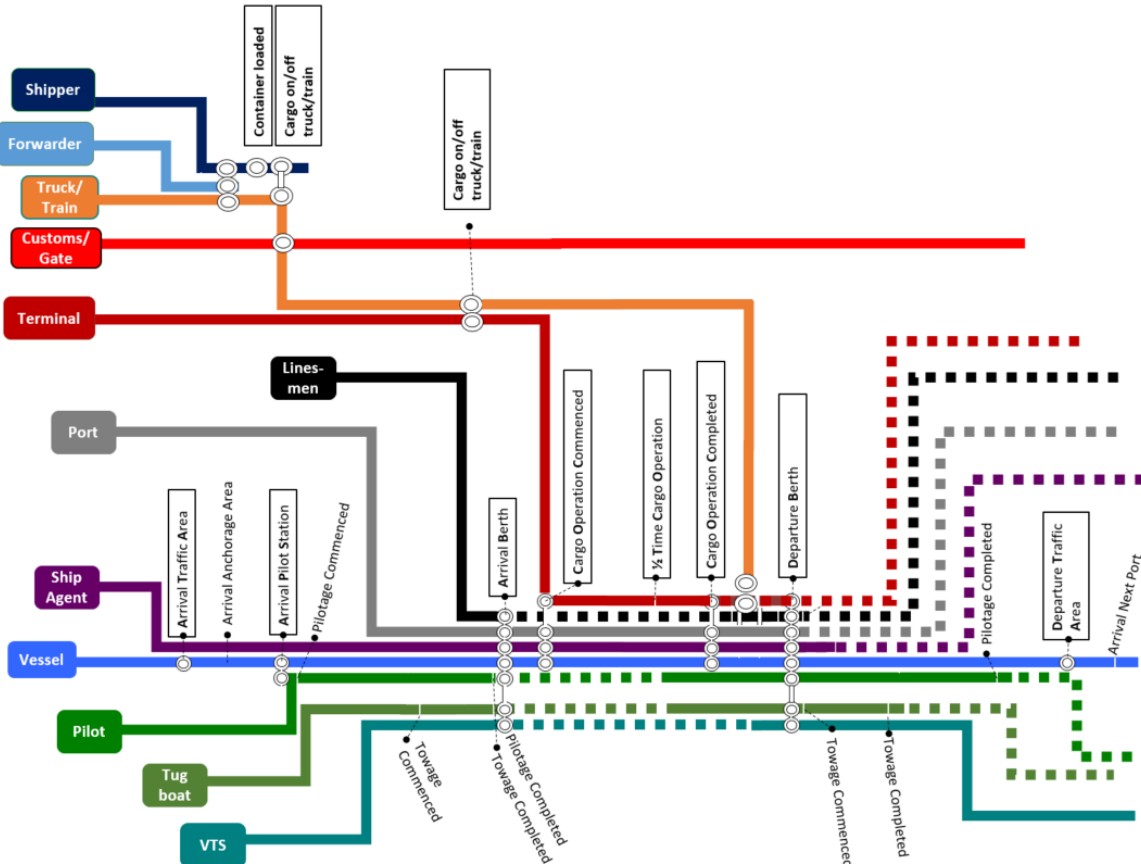

**Figure 3.** Port Activity Application Metromap. Metromap highlights the complexity, and the need for collaboration between multiple port call actors, in staging a port call.

As can be seen in Figure 3, there were many states in which the actions of several actors need to be synchronized. States associated with the vessel's turn around process was also considered as the common object of interest among participating actors. Efficient resource allocation and port call execution required predictable state changes and would therefore benefit from improved communication and collaboration among participating actors. The Port Activity Application aimed at being a platform enabling such support.

States could also be used as waypoints for actors to plan and monitor the status of the process for an increased awareness and ability to predict upcoming events. The states selected for EfficientFlow (Port of Gävle and Port of Rauma), are all events that could be reported (either automatically or manually) in the Port Activity Application module and visualized in the image for situation awareness that was shared among the key port call actors involved in the port calls in Gävle and Rauma.

In order to briefly explain what the enhanced business model entails, a business canvas tool was used. The business model, shown in Figure 4, gives better understanding of what the business logic was and how the business model influenced the end customers.

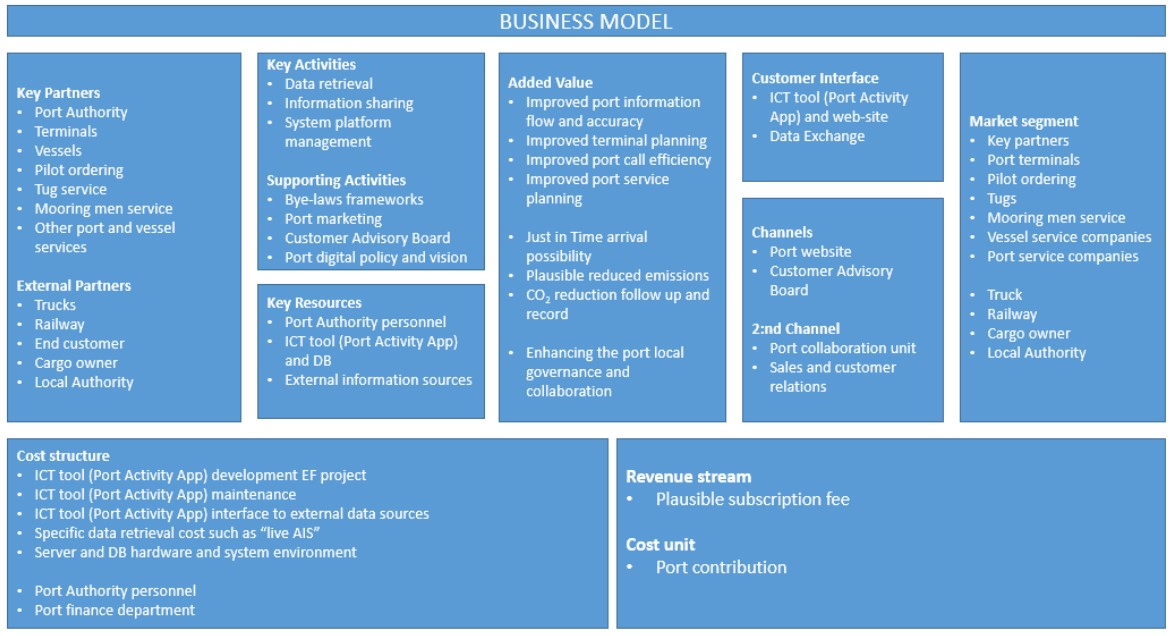

**Figure 4.** Business model to highlight the business logic and influences of the business model on the end customers.

The business model was planned to save time by optimizing the operations of the whole supply chain. The target market involved at least the following actors: Key partners and external partners. The main customer interface with port call actors and terminals was covered by the Port Activity Application (PAA). Data exchange was available for key partners. The PAA was accessible from any type of device. The end-users had continuous access to a real-time updating database due to an automated machine-to-machine communication. A Customer Advisory Board (CAB) was also established. The port organisation and port call actors focused on efficiency and safety, on one hand, and on the other hand, they focused on quality including design, brand status and end customer satisfaction. The key activities generating added values was data retrieval and information sharing. Another important activity was the management of the IT platform and IT environment of the PAA. A very important supporting activity was the Customer Advisory Board meetings. The purpose of the Customer Advisory Board, as mentioned before, was to collect information from different port actors and their procedures. Port call actors were considered as key partners. Key partners and external partners were interconnected together in the real-time information sharing integrated system. Key Resources, include the ports taking responsibility to maintain and develop PAA, either as part of the port

organization and as an outsourced service. The qualification and the motivation of the personnel as a fundamental resource was kept as high as possible. Another vital factor was the key partners' contribution. The integration of different systems and the continuous information flow have always been maintained in good order. Revenue Streams, for example, plausible subscription fee for port call actors or port contribution could be used. The business model reflects a partly idealized picture of port operations. During the implementation of the business model, following aspects must be taken into account: The current environmental factors such as weather conditions (heavy winds, storms, waves, ice etc.) in the Baltic Sea region, climate changes as a strategic perspective and in terms of existing environmental protection strategies; and operational factors of the port system, such as changes in local and general legislation, strikes or force majeure circumstances and how this may affect the arrival/departure time of the ship, for example.

The enhanced business logic required front-end PAA. The PAA is a collaborative platform where the port call actors could interact with each other. The application allows monitoring of cargo, transportation and information flow as well as following the port facilities and the status of port services.

The ultimate goal was to link and keep all the available information visible and accessible at all times to all parties involved in the port call process, so they can benefit of it in their decision-making and planning processes. Therefore, the user interface and the data exchange logic were agreed with the end users during multiple joint workshops and interviews during and after testing process.

### 3.3. Information Flows for Intermodal Sharing of Information

A clear finding of the auditions was that the hinterland logistics operators had little interest in the Port Activity App as it was. In relation to arriving shipments, the stakeholders had some interest in estimated time of arrival (ETA) and actual time of arrival (ATA) information of ships provided, but this information was also generally available in the web pages of the ports and thus using the application for this did not create much value.

When considering outbound shipments, the Information on estimated time of departure (ETD) or actual time of departure (ATD) of a vessel provided by the application did not have much relevance for operational nor planning purposes of hinterland operators either, as cut-off time for delivering a container to the port is generally several days before the departure of the vessel. The most critical factor in shipping containers outbound at Rauma port was also related to the availability and booking of the container as the booking process currently is manual and needs to be done by e-mail or phone.

As most of the operators served many ports, one challenge identified was also that each port has their own application. There was, but a genuine interest in developing the digitalization and automatization supporting the connectivity of hinterland and port operations further. To add further value for the hinterland operators, relevant Estimation to Complete (ETC) information related to the arriving container status should be added to the application. Information related to the availability of empty containers should also be digitalized and provided in an accessible format. This information was available in digital form in the operational systems of parties operating in the port but not yet directly visible or accessible for external parties.

The result of the auditions in Sweden supported the findings done in Finland. Hinterland logistics operators had little interest in the Port Activity App. The stakeholders in Sweden had some interest in estimated time of arrival (ETA) and actual time of arrival (ATA) information of ships provided but this information is also generally available in the web pages of the ports and thus using the application for this would not create real value.

As in Finland, the information on estimated time of departure (ETD) or actual time of departure (ATD) of a vessel is not very relevant for operational nor planning purposes of hinterland operators as cut-off time for delivering a container to the port is generally several days before departure of the vessel. The most critical factor in shipping containers outbound at the Port of Gävle was also related to the availability and booking of the

container as the booking process currently was manual and needed to be done by e-mail or phone.

As the information sharing process currently was quite manual also in Sweden, there was a genuine interest from the hinterland operators to develop the digitalization and automatization supporting the connectivity of hinterland and port operations further. As in Finland to add further value for the hinterland operators, relevant estimation to complete (ETC) information related to the container status should be added to the application.

The railway operator in Sweden was very interested in developing data sharing. This would greatly help the planning of logistics operations both inbound and outbound. As the volumes with railway transportation were generally quite big, the need for information supporting planning well beforehand would be welcome. When collecting containers from the port, further information on transportation capacity is required (weight/volume) and unloading sequence to plan the setup of trains and wagons would support smoother operations.

The port operator in Gävle had data on unloading or availability of container for pickup by the trucking company available in their terminal operating system and is working on extending the data to include estimated time of completion (ETC) information.

### 3.4. Testbed

To provide a deeper analysis of the cycle of operations in the port, the "Operations and associated times of maritime logistics scheme" [55] (p. 74) was adopted for the Port of Rauma. Figure 5 shows the time and place that an operational activity takes in regard to other activities. At the same time the scheme makes it possible to track bottlenecks with an increased risk of losing time due to the load intensity at a particular stage.

The challenges that were faced during the development of the applications, included the use of various IT systems; multiple points of unaligned information with no or limited data transfer; vast amount of information and communications back and forth between different actors; no centralized place or access to reliable information; no working warning system or ability to react to unexpected events All these factors result in the planning horizon being more limited.

The purpose of the testbed was to present the test methods for different test groups, i.e., the tasks and goals each test group have had, the achieved results, and finally the analysis and evaluation of the results. The scope of the tests was to locate the most critical bugs, improve the applications user-friendliness and to optimize the servers, so they would not have any misfunctions, even if there was tens or hundreds of simultaneous users. At the time of starting the test process, the software for both ports was in a "minimum viable product" (MVP) phase. It had all the necessary components, such as integrations between the application and different port operation systems, Activity page and a possibility to send and receive ship-specific notifications to all related parties. Port Activity Rauma also had functions for following the logistic flows in the port.

The first test with the student group of SAMK information technology classes focused at the Port Activity Rauma's user interface and user experience.

During the test by the SAMK experts, which was ongoing for several days, the test users logged in to the application to cross-check the data that was shown in the Activity page. At the same time, the test users had to check many other port operation systems where the data was coming to the application from and see how accurate the data was. Findings were collected into a separate Microsoft Excel sheet with proper information and screenshots of the erroneous functionality of the app. The data was forwarded to the software development company. During the testing and fixing process, an accuracy of the system improved step by step until it was ready for the testbed with actual port actors.

Tests in both ports, the Port of Gävle and the Port of Rauma, were more practical and involved more port actors and tasks in a real-life environment than originally aimed at. Valuable information, feedback and new ideas were collected regarding the future development of the applications—e.g., truckers liked to see when their container was ready to be picked up, a tug company liked the possibility of sending invoices from the app and

the possibility to integrate a map-view within the apps Activity or ship-specific view. Most of the port actors were delighted about the simplicity of the app and found it very helpful compared to the traditional way of communicating everything through a phone call.

Overall, the comprehensive testing with various parties and partners gave great insights and great advantage for port actors. Everyone involved in the testing was pleased to be part of building something that will potentially make a huge impact in the whole industry. The current status of the application provided the opportunity to get closer to the targets of the EfficientFlow project.

Interview situations were separate from each other, and respondents were not aware of the responses provided by others. As an over-arching finding from the interviews was the contribution the application can provide for the port calls not only during the period of navigation but while moored and planning the departure. The Port Activity Application includes detailed information on how the traffic is arranged and what the future looks like in navigable waters near the port. This assists in planning and scheduling the departure of a safe outbound voyage. Moreover, when the schedule is clear and foreseeable, stevedores, linesmen, pilot etc., can be arranged in good time without the risk of charging for stand-by hours.

By comparing the answers of the interviews, conducted at Port of Gävle in 2018 and in 2020, it was observed that more than 50% of respondents agreed that when in the Port Activity App information is displayed, such as Live ETA, i.e., current times when ships are at the quay, ETD etc., it will certainly decrease phone calls. Respondents also agreed that it was useful for their daily planning tasks and increased efficiency when in the PAA information about a ship call to the Port of Gävle was gathered in one place and was available in real time, around the clock. Almost all respondents shared the opinion that communication misses and sharing outdated information decreased when the Port Activity Application was available for all users. In addition, almost all respondents acknowledged that it is an advantage that the information in the Port Activity Application is automatic and largely not dependent on the person entering it when they had the time and opportunity and that the 24/7 availability of the Port Activity Application and automatic data entering would make it possible to trust the information better. After testing the Port Activity App in real life, most of the respondents agreed that a special function for sending out push notices to all port actors would improve the ability to convey information in case of a sudden event or delay. As the answers of 2018 were compared with the results of 2020, a positive view can be seen in the respondents' opinion related to the new Port Activity App.

General comments mentioned regarding the Port Activity Application and its functions as well as the upcoming features such as sea chart and queuing systems (just-in-time) from the behalf of port actors were positive: "Nice feature with sea chart but may take some time to acclimatize to new procedure"; "Vessel planned berth number was good information and very useful", "Automatic Identification System (AIS) vessel position on a sea chart was very useful feature", "Queue system was a very interesting feature that will improve terminal efficiency as well as reduce negative impact on the climate". The comments of the questionnaire participants showed positive feedback and growing popularity of Port Activity Application among main users and stakeholders. The result proved that the expectations of the respondents had been met. The results of the questionnaire 2020 defined that the Port Activity Application with automated and online information flow had a significant impact on the information flow efficiency. The Port Activity Application is also a springboard to improve the work procedure for port actors which had a positive impact on the port flow process.

The screenshots on Figures 6 and 7 display an example of the information provided by the final version of the Port Activity App developed during the EfficientFlow project and currently in a use in real life by Port of Rauma, and Port of Gävle, respectively.

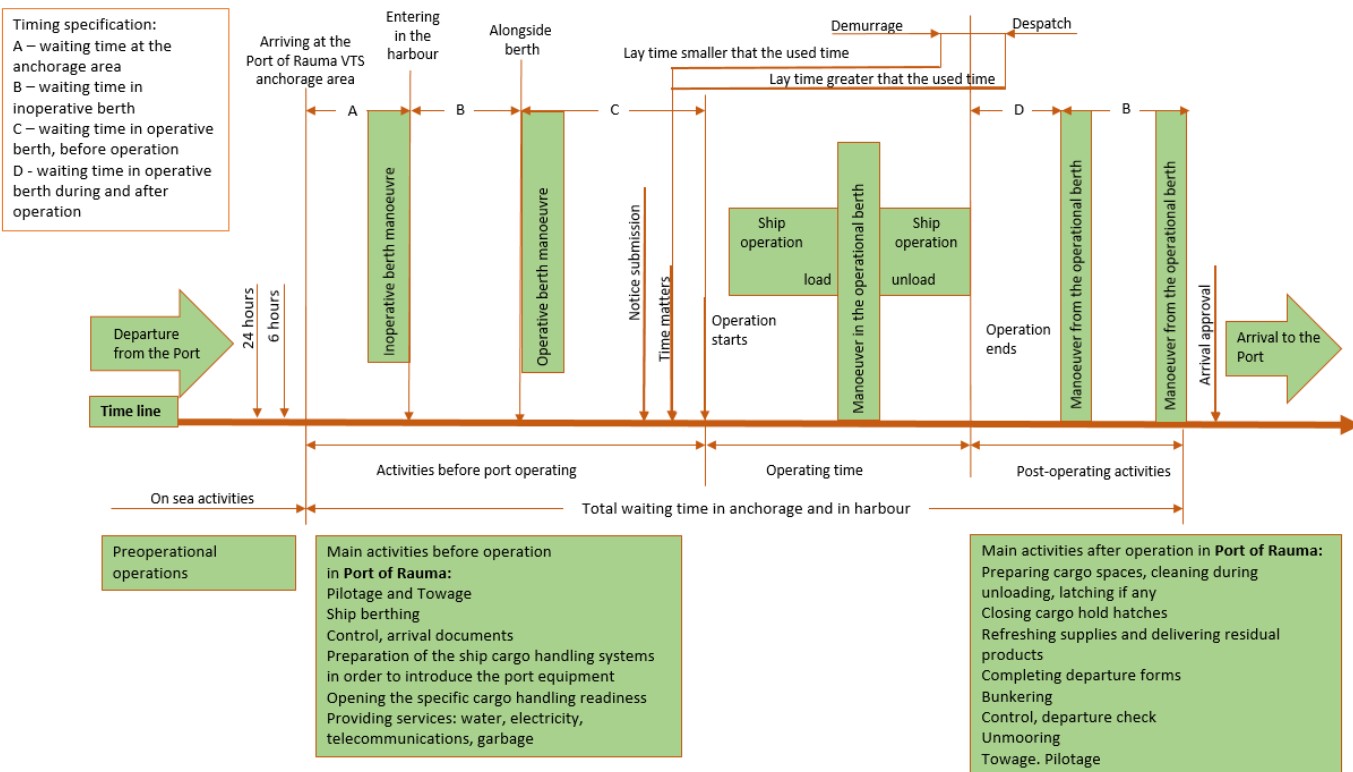

**Figure 5.** Main port operations with timelines (Port of Rauma, Finland).

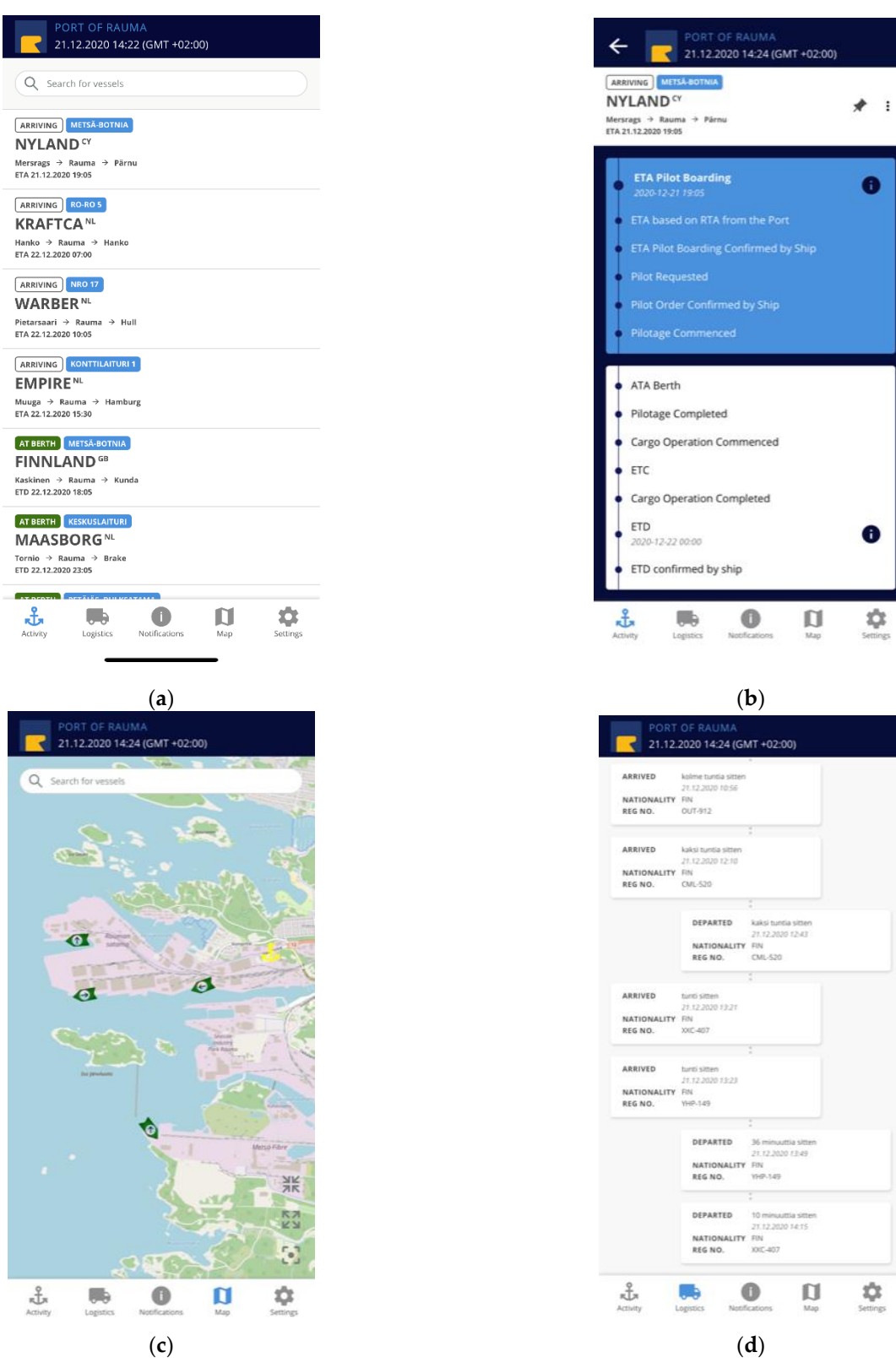

**Figure 6.** Screenshots from final version of the Port of Rauma mobile application: (**a**) The main page where active ships are shown; (**b**) The ship specific page displaying different time stamps; (**c**) The port specific map of Port of Rauma; (**d**) The logistics page with information related to truck movements.

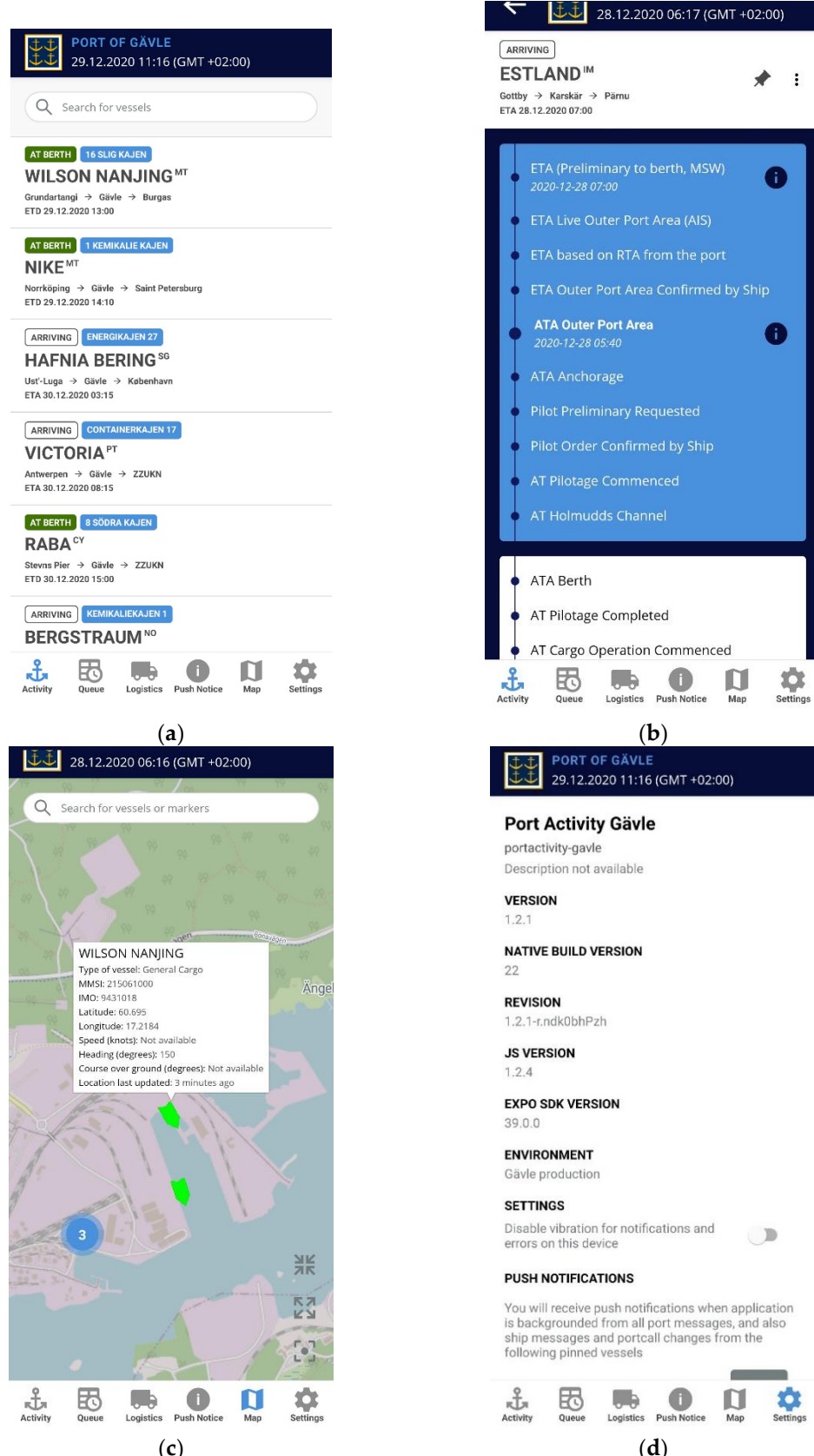

**Figure 7.** Screenshots from final version of the Port of Gävle mobile application: (**a**) The main page where active ships are shown; (**b**) the ship specific page displaying different time stamps; (**c**) the port specific map of Port of Gävle; (**d**) The settings page with version information etc.

## 4. Discussion

### 4.1. The Business Model Implementation, Challenges and Possibilities for Port Actors

One of the recent texts mining research of 155 publications on the maritime sustainable studies has shown that the main issues are related to the sustainability of ports and shipping, carbon emissions and regional environmental regulation, route optimization and lower logistics costs and supply chain management [56]. Until now, existing publications are limited and do not cover all aspects related to sustainability in maritime logistics and shipping [57]. This research delivered improved processes, business models and ICT tools for enhanced information exchange between port actors, between ports, between port and hinterland operators and between ports and ships.

The overall goal with the Port Activity Application was to enable the sharing of intentions and actual state updates among involved actors to enable enhanced decisions concerning each actor's coordination of the upcoming operations to be performed. The Port of Rauma and the Port of Gävle presented the first implementation of the Sea Traffic Management concept in the Baltic Sea Region. The results of the project serve as examples of best practices for other corridors and ports in the Central Baltic area and beyond. Improved information exchange between the actors of the two corridors in the Central Baltic area contributed to higher efficiency, just-in-time approach and timesaving, reductions in transport time, higher transport predictability and improved sustainability and transport quality.

The maritime sector is highly dependent on cleantech solutions and their implementation. HELCOM's 2018 report on the state of the Baltic Sea describes eutrophication, plastic pollution and the impact of the global climate as additional stressors for the region [58]. Maritime transport intensity is increasing, and new sustainable solutions related to reducing $CO_2$ and other emissions are needed for the industry.

The transition to circular economy, for example, begins with the creation of an appropriate business model, which takes into account the interaction of various services and, for example, a decrease in the part of port cathedrals in the overall profit structure [59]. The smart port concept faces also the challenge of sustainable blue growth, which can be addressed by improving the efficiency of shipping and supply chain management, including port calls and planning cooperation with port and hinterland operators [60].

The operating environment of a seaport is a complex system of interaction between physical components, socio-economic conditions and ecological systems [61]. "Digitalization and digital business are inevitably shifting operational and management focus of organizations towards being business ecosystems. Business ecosystems are the dynamic network of entities (people, businesses and things) interacting with each other. Digital business has gained momentum in all commercial sectors—and maritime transportation cannot be immune to these changes—indeed the changes are happening already" [6] (p. 12). For example, the "smart port concept can be displayed as the port where the environmental impacts, operations, and the energy consumption are addressed" [60] (p. 92). A smart port creates an environment capable of ensuring the safety, reliability, environmental friendliness and efficiency of its members [60].

Each stakeholder participating in the logistics chain can benefit from using the developed results of this research and the Port Activity App. For almost all port/terminal operators during the off-peak periods, overmanning is a very common issue. By being aware of the cargo flow, in advance, by applying the enhanced business model in practice, the actors are be able to predict and plan the expected workload. As a result, the terminal operators save time and costs due to better long-term and short-term planning, improved labour management and less cargo handling operations. Some other areas, where the Port Activity Application can improve the port operator's performance, due to more accurate estimated times, include capacity planning, cargo flow scheduling, dwell time and cargo transportation. This undisputedly leads to significant time savings as well.

One of the problems of the smart port is also the problem of intermodality, which requires technological innovations on ships and in ports, including those related to energy

efficiency and CO$_2$ emissions [60]. So far, the shipping lines have received the actual cargo data, stowage plan and all cargo data before the ship's departure. Now shipping lines can monitor the cargo handling operations in real-time. Another advantage is the real-time updates of the berth's schedules. This results in the possibility of adjusting the ship's steaming speed according to the latest berth status. An important step forward is the recently-adopted STM contractual clause by BIMCO [62], which will permit further improvement in the efficiency of the ship and port operations.

By using the Port Activity Application, the port authority and port services have an excellent overview of the ships', hauliers' and cargo traffic in the port area. The carriers' and the port's operational plan can be aligned. Port service providers, such as mooring services, tugboats, icebreakers, pilots, etc. can be arranged adequately in due time. Pilots can be automatically notified early enough in case of change in the ship's schedules. The land transportation companies can plan their resources and optimize their operations efficiently. The hauliers can schedule optimal trips according to the cargo type and the cargo volume avoiding idle runs between two consecutive shipments. The automated port access can save extra time if the port gate is integrated with a Port Activity Application.

The benefit for the industry in general lies in the fact that the delivery time depends on many factors, but one of the most variable, and thus, important factors is the time of the cargo staying in the port area. The ports offering shortest transit times are more competitive than the others. The transit time, however, includes the time needed for obtaining customs clearance. Therefore, the enhanced business model promotes excellent collaboration and information sharing between customs, port authorities and terminal operators, in order to improve the information flow and to reduce the total lead time. The Port Activity Application can significantly reduce the workload of the ship's agents. Instead of being the middle-man, the enhanced business model can assign them supervisory duties. The new role as a second control level responsible person can allow the agents to coordinate the information and cargo flow better. The region around the port can gain ecological benefits due to optimization in transportation planning and vehicle management. Less idle movements and shorter port calls will significantly help meeting the transport emissions targets set by the European Commission in its White Paper on Transport; a 20% reduction from the levels of 2008 by 2030, and a 60% reduction from the levels of 1990 by 2050. The harmful emissions from international shipping are to be reduced by at least 40% from the levels of 2005 by 2050 [63].

*4.2. Port Activity Application, Digitalisation Processes and Port and Hinterland Actors' Cooperation*

Technological solutions and innovations aimed at the sustainable development of the industry are highly dependent on cooperation between stakeholders at all stages of implementation [64]. Functional Local Governance results to an organizational framework by setting internal policies and practices that provide guidance and instructions. The Local Governance also models organization structure, revealing necessities for internal by-laws in accordance with the domestic and international standards, rules and regulations aiming to govern the established partnerships. The Port Activity Application contributes to the situation with the new EU directive [44]. New port call actors' (private companies) introduction in the ports can be smoother with the Port Activity Application in place. From the overview of both Local Government Structures (LGS) it could be concluded that a positive influence and a competitive advantage could be gained regardless of the applied model if the approach is in line with the organizational goals of all port actors and it is built around the partners' development strategies. The sustainable development of the region in which the port operates should be included in the planning for the sustainable development of the port, including and maritime logistics with transport services and hinterland supply chains [56].

According to the research, the most important function of the Terminal Operating System is tracking the arrival and departure times of both ships and land transport. In this regard, it is important for the functioning of the system to have all the necessary maximum

relevant information [65]. Intermodal sharing of information is crucial when managing intermodal logistics. Digitalization can typically be initiated at the port by managing and automating information flows within the port and between the port and the vessels. The development of the Port Activity application was definitely a prominent step forward on the Port digitalization roadmap. From the view of managing the whole intermodal flow efficiently, this was a good start—to continue, the application needs to be extended to managing information flows with the hinterland and between ports connecting all the logistics operators handling the goods, and other stakeholder including both shipping parties. It is vital to ensure efficient information sharing between operators within the intermodal logistic chain, in order to achieve the potential benefits of this transportation method. Intermodality provided the possibility to choose the most effective method of transportation in respect of economics, lead time, reliability, and time management. The effectiveness of all these factors was affected by correct and timely information being available to all operators within the delivery chain. During the investigation it was observed that centralized and automated information exchange methods in the scope of the research had been minor before implementing the Port Activity Application developed by the project. Methods of gaining crude or precise information varied from operator to operator. The potential to develop information sharing to support operations has however been identified and activities to digitalize and automate operations are ongoing and being planned further.

Within the frame of logistics, in this research, information was still fragmented to several locations, the methods for gaining information were numerous and information sharing occurred mainly between neighbouring operators in the supply chain. Predictability would be increased if information was available from two or three nodes for allowing more time for the operators to adopt in changes in timetables or other affective occurrences. Studies showed that to gain optimal benefit from intermodal transportation, end to end information of the current state of the container should be readily available, as well as the information of circumstantial deviation leading to changes in the activities further in the supply chain [10,11,47–51]. These were also items that were raised by the interviewed actors as needing development. The current state of the container, includes geographical locational data of the container, forwarder, transportation method, vehicle, location of the container on the ship, neighbouring containers, and unload sequence.

This study did not specifically calculate the cost savings of the port and port operators. However, future studies can draw attention to this, as, for example, in studies on the creation of a business model that allows to combine cargo flows of neighbouring ports to fill a regular rail or barge service, as it is possible to significantly reduce both internal and external costs of transport companies and port operators [66].

For proper functionality of the Terminal Operating System it is very important to keep updated of loading lists for terminal operators, with information about the cargo, including information about its location [65]. Currently, data regarding the container or other cargo for transportation is not easily available, it is not always precise and methods for gaining this data varies. Intermediate information sharing between nodes needs to be further developed. "Intelligent container technologies (RFID) and real-time tracking of cargo increase the transparency of the transport route from the sender to the recipient. Shipping companies will be able to use their own tracking applications in the near future, in which the location of the container can be determined by using a GPS signal. Due to the use of modern sensor chip technologies, a large amount of data is already recorded at sea and analysed onshore, which allows optimizing technological flows on board, as well as during processing in ports. It also reduces waiting times and costs" [67].

For instance, a container is shipped from Gävle to Rauma by a sea vessel and at Rauma port the following trucking forwarder takes over the transportation from Rauma port onwards. The trucking forwarder has semi-reliable information of which sea vessel the container is loaded on, what the estimated time of departure (ETD) and estimated time of arrival (ETA) are. A final confirmation of the loading is not updated nor deviations in

ETA due to weather, or technical breakdowns or other variables affecting the transportation schedule. Forwarders gather information on cargo status with differing methods, which are often manual, inefficient and inaccurate. As the shipment status not being automatically updated in a centralized fashion, a forwarder relies on very traditional methods, including person-to-person telecommunication combined with manual searches in databases. This consumes time and resources, which was also identified by these forwarders. Forwarders also rely on information based on causal experience and third-party information. For instance, a trucking company might interpret weather forecasts to judge if the ship is able to dock in schedule or not. Decisions are made by the captain on the vessel, but information is not distributed to the supply chain to parties impacted by such decisions. Approximative unloading schedules of containers from ship to harbour are not available, estimates are made based on the weight of the container to be forwarded. The PAA also has a Logistic module providing gate timestamp information on the trucks. This could possibly be developed further to provide data also related to the containers/cargo status.

Other important functionalities of the Terminal Operating System include emergency reports, retrieve and dispatch requests for gate operations, allowing the customization of the layout, and providing extended data information about vessel voyages (e.g., security requirements or maintenance information) [65]. Regardless of how advanced and sophisticated the system, it cannot react adequately in exceptional situations and on unexpected events. Therefore, an option for manual inputs could be provided. This benefits not only better monitoring and control, but also proactive intervention in case of operational disturbances. The integrated warning system in case of unexpected events is available to everyone. By creating and responding to manually created entries, the participants will be able to avoid or at least to reduce the impact of any undesired events on their schedules and arrangements. This option is crucial for the efficient workflow, time and cost savings. For the proper work of the warning system a control body is required: the first level control is executed by present Application Programming Interface logic, and the second level of control is performed by designated persons. In future port applications, also the problem of the first-last mile has to be taken into account, such as firs-last mile issues in the integration of passenger and freight transport in terms of combining passenger and freight traffic, as well as related changes in the business models of public transport and the supply chain in general [68] or for example, issue what was touched upon in studies related to the environmental footprint of land transport [69]. In the first-last mile system evaluation and planning study, improved transport routing planning was established as potentially being one of the solutions for reducing general costs associated with first-last mile processes [70]. In our pilot ports, there was only cargo traffic, so there was no first-last mile issue. However, when using PAA in ports with more intensive and especially mixed cargo/human traffic, this model has to be adjusted to local circumstances to interpret the local business environment.

Each country uses its own system for collecting and delivering the data and its own Maritime Single Window (MSW, but PAA could be valuable for the middle or big size ports in other countries for use as digital tool for the port operations planning). The future in the development of the Single Maritime Window system should be directed towards full harmonization of interfaces that will be available to all EU operators and towards standardization of all maximum data available to the port to ensure true dispatch only once [8]. Port operators both in Gävle and Rauma have proved that the developed Port Activity application has had tremendous effects in improving efficiency in the ports. Some hinterland operators also stated that some of the information currently available with the application supports their operations and most of them would be very interested in future development possibilities of the application and other PCS solutions supporting port and hinterland digitalization and automatization. Based on the research findings and results of the project, major development and steps in digitalization have been achieved in the development of the Port Activity Application (PAA) [71]. Fintraffic takes ownership of the developed application in Finland in the future, and proves credibility of the application and

secures the future evolvement of the application and port digitalization. Interest towards the application in the beginning of this project was huge by different ports and by Fintraffic. Finally, in November 2020, Fintraffic and software development company agreed to offer the open-source platform as a Software-as-a-service (Saas). Fintraffic sees that a natural ecosystem and a community of developers can be built around the application thanks to the possibilities of the open source solution. Each port can use the timestamps that are freely available, or use the application by themselves and develop the application to their specific needs with a chosen partner.

**Author Contributions:** Conceptualization, O.d.A.G., H.K., J.M.M. and M.M.K.-T.; data curation, H.K.; Formal Analysis, O.d.A.G., H.K. and J.M.M.; funding acquisition, H.K. and M.M.K.-T.; Investigation, O.d.A.G., H.K. and J.M.M.; methodology, O.d.A.G., H.K., J.M.M. and M.M.K.-T.; Project administration, J.M.M., H.K. and M.M.K.-T.; resources, H.K.; software, H.K. and J.M.M.; supervision, M.M.K.-T.; validation, O.d.A.G., H.K., J.M.M. and M.M.K.-T.; visualization O.d.A.G., H.K. and M.M.K.-T.; writing, original draft preparation, O.d.A.G., H.K., J.M.M. and M.M.K.-T.; writing, review and editing, O.d.A.G., H.K., J.M.M. and M.M.K.-T. All authors have read and agreed to the published version of the manuscript.

**Funding:** This research was funded by EU (European Regional Development Fund, Interreg Central Baltic), project EfficientFlow (CB_607). The publishing fee was supported by The Baltic University Programme.

**Institutional Review Board Statement:** Not applicable.

**Informed Consent Statement:** Not applicable.

**Data Availability Statement:** The data presented in this study are available on request from the corresponding author. The data are not publicly available due to business related information of the pilot cases. The source code for the Port Activity Application is released under an open source license. This data can be found here: https://port-activity.github.io/.

**Acknowledgments:** The authors are grateful to the experts from Swedish Maritime Administration, Satakunta University of Applied Sciences, Port of Rauma, Port of Gävle and Traffic Management Finland on contributing to data collection. They also express their thanks to M.Sc. Hanna Rissanen for grammatical advice on the article.

**Conflicts of Interest:** The authors declare no conflict of interest.

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
