# Peer review of "Digitalization in Just-In-Time Approach as a Sustainable Solution for Maritime Logistics in the Baltic Sea Region"

_sustainability, doi:10.3390/su13031173_

Round 1

Reviewer 1 Report

I suggest for Authors add of the problem of climate changes. They write about it at the beginning, line 33. In the business model, fig 4, isn't any connection with the climate changes. The climate changes have connection with number and force of storms on the Baltic Sea. It is directly connected with the sea transport , fig. 2. So, I suggest to add "elements" of the weather to the Business model, fig. 4. These "elements" of weather may delay of planned time of the sea transport on the Baltic Sea. In the business model is model of ideal situation. So, what can be happen with planned time of operation for one customer, if on Baltic Sea will be long time strong storm, for example 3 weeks ? Baltic sea is closed shallow and very small.
Another case. Authors in abstract write about reduction emission of the CO2 gase. In paper isn't any results of tests of usage of LNG gas for powerig the ships. On 6 lines, 732-737, in Discussion are little words about it. No in the main text of paper. Better is remove this part.

Author Response

Point 1:

I suggest for Authors add of the problem of climate changes. They write about it at the beginning, line 33. In the business model, fig 4, isn't any connection with the climate changes. The climate changes have connection with number and force of storms on the Baltic Sea. It is directly connected with the sea transport, fig. 2. So, I suggest to add "elements" of the weather to the Business model, fig. 4. These "elements" of weather may delay of planned time of the sea transport on the Baltic Sea. In the business model is model of ideal situation. So, what can be happen with planned time of operation for one customer, if on Baltic Sea will be long time strong storm, for example 3 weeks? Baltic sea is closed shallow and very small.

Response 1:

Thank you for this comment. We have added weather to Business model description as well as other circumstances affecting the implementation of the business model (lines 541-548 in the track changes version). In addition, in introduction the theme is clarified within a couple sentences (lines 32-41 in the track changes version).

Point 2:

Another case. Authors in abstract write about reduction emission of the CO2 gase. In paper isn't any results of tests of usage of LNG gas for powering the ships. On 6 lines, 732-737, in Discussion are little words about it. No in the main text of paper. Better is remove this part.

Response 2:

Thank you for this remark, LNG gas part is deleted (lines 773-776 in the track changes version).

Reviewer 2 Report

The paper “Digitalization in just-in-time approach as a sustainable solution for maritime logistics in the Baltic Sea Region” has been submitted for publication in the journal Sustainability [Manuscript ID: sustainability-1075348]. It is an interesting manuscript, dealing with the coordination and information exchange mechanisms between the existing systems of ports and ships, also including sustainability issues – particularly addressing CO2 emissions. The contents belong to the core business of this journal, and the manuscript reads smoothly throughout.

I feel that we are dealing with an innovative research, that could further be improved in the following points:

  1. A general lack of consideration of the port accessibility as a whole has been given in this paper, as the authors do focus on the ports only. I do not feel like the model should completely be changed. Nonetheless, it would be very useful to add a paragraph somewhere, addressing this issue and explaining why it has not been included in the questionnaire. Recently, some good materials on the first-mile last-mile problem have published in the literature [Francesco Bruzzone et al., The integration of passenger and freight transport for first-last mile operations. Transport Policy Volume 100, January 2021, Pages 31-48; Silvio Nocera et al. How to evaluate and plan the freight-passengers first-last mile. Transport Policy, Available online 8 January 2020, doi: 10.1016/j.tranpol.2020.01.007; Edoardo Croci et. al. An LCA comparison of last-mile distribution logistics scenarios in Milan and Turin municipalities. Case Studies on Transport Policy, Available online 9 December 2020, doi: 10.1016/j.cstp.2020.12.001];
  2. Both sections regarding testbed (2.6 and 3.4) are definitely too long and someway redundant. I would recommend the authors to cut them by a good third;
  3. The same applies to the discussion section – which is definitely too long in the present version. The authors should consider the perspective to divide it in two or more sections, or to cut it consistently;
  4. The model does not include any primary pollutant, whose social impact deriving from logistics is as relevant as the one deriving from carbon dioxide emissions [See Elnaz Irannezhad et al. The effect of cooperation among shipping lines on transport costs and pollutant emissions. Transportation Research Part D: Transport and Environment Volume 65, December 2018, Pages 312-323; Eva Merico et al. Development of an integrated modelling-measurement system for near-real-time estimates of harbour activity impact to atmospheric pollution in coastal cities. Transportation Research Part D: Transport and Environment Volume 73, August 2019, Pages 108-119]. Yet, no explanation about this peculiar choice can be found in the paper. This is particularly relevant, considering the choice of the journal made by the authors;
  5. The authors should indicate the margins of generalization of such an interesting research line.

Author Response

Point 1:

A general lack of consideration of the port accessibility as a whole has been given in this paper, as the authors do focus on the ports only. I do not feel like the model should completely be changed. Nonetheless, it would be very useful to add a paragraph somewhere, addressing this issue and explaining why it has not been included in the questionnaire. Recently, some good materials on the first-mile last-mile problem have published in the literature [Francesco Bruzzone et al., The integration of passenger and freight transport for first-last mile operations. Transport Policy Volume 100, January 2021, Pages 31-48; Silvio Nocera et al. How to evaluate and plan the freight-passengers first-last mile. Transport Policy, Available online 8 January 2020, doi: 10.1016/j.tranpol.2020.01.007; Edoardo Croci et. al. An LCA comparison of last-mile distribution logistics scenarios in Milan and Turin municipalities. Case Studies on Transport Policy, Available online 9 December 2020, doi: 10.1016/j.cstp.2020.12.001];

Response 1:

Thank you for this suggestion.  The aspect of the first-last mile problem and relevant references are added to the manuscript (lines 965-976 in the track changes version).

Point 2:

Both sections regarding testbed (2.6 and 3.4) are definitely too long and someway redundant. I would recommend the authors to cut them by a good third; 

Response 2:

This was a very welcomed remark. Indeed, the sections were a bit long. Both of the sections have been cut and main aspects highlighted (see the track changes version of the manuscript under 2.6. and 3.4.)

Point 3:

The same applies to the discussion section – which is definitely too long in the present version. The authors should consider the perspective to divide it in two or more sections, or to cut it consistently;

Response 3:

Discussion section has been cut and subtitles have been added. (see the track changes version of the manuscript)

Point 4:

The model does not include any primary pollutant, whose social impact deriving from logistics is as relevant as the one deriving from carbon dioxide emissions [See Elnaz Irannezhad et al. The effect of cooperation among shipping lines on transport costs and pollutant emissions. Transportation Research Part D: Transport and Environment Volume 65, December 2018, Pages 312-323; Eva Merico et al. Development of an integrated modelling-measurement system for near-real-time estimates of harbour activity impact to atmospheric pollution in coastal cities. Transportation Research Part D: Transport and Environment Volume 73, August 2019, Pages 108-119].  Yet, no explanation about this peculiar choice can be found in the paper. This is particularly relevant, considering the choice of the journal made by the authors.

Response 4:

This is an important remark and we fully agree that sustainability has components of environment, economics and social aspects. However, in our manuscript we have concentrated on environmental and economical aspects as their interlink to IMO regulations and European Union Strategy, focusing especially on reducing CO2 emissions as main driver of the climate change. In future studies, we acknowledge high need on combining social sustainability within logistics chains. This needs a vast cooperation of experts from various sectors such as maritime logistics and social scientists.

Point 5:

The authors should indicate the margins of generalization of such an interesting research line.

Response 5:

The aspect if first-last mile problem and relevant references are added to the manuscript in lines 965-976 in the track changes version.

Round 2

Reviewer 2 Report

Paper improved a great deal in this revision round. Ready for publication